**Data Availability Statement:** Our data cannot be shared publicly because they contain sensitive participant information (i.e., personal interviews

# How can HIV self-testing facilitate increased access to HIV testing among multiply marginalised populations? Perspectives from GBMSM and trans women in England and Wales

**Isaac Yen-Hao Chu**[1,2]*, **Fiona M. Burns**[1,3], **Talen Wright**[4], **Phil Samba**[5], **T. Charles Witzel**[1,2,6], **Emily Jay Nicholls**[1], **Leanne McCabe**[7], **Andrew Phillips**[1], **Sheena McCormack**[7], **Alison J. Rodger**[1,3], **Peter Weatherburn**[2]

1 Institute for Global Health, University College London, London, United Kingdom, 2 Faculty of Public Health and Policy, London School of Hygiene and Tropical Medicine, London, United Kingdom, 3 Royal Free London NHS Foundation Trust, London, United Kingdom, 4 Bipolar UK, London, United Kingdom, 5 The Love Tank CIC, London, United Kingdom, 6 Faculty of Social Sciences and Humanities, Centre of Excellence in Research on Gender, Sexuality and Health, Mahidol University, Nakhon Pathom, Thailand, 7 Medical Research Council Clinical Trials Unit at University College London, London, United Kingdom

* isaac.yh.chu@ucl.ac.uk

## Abstract

### Background

HIV self-testing (HIVST) may facilitate marginalised populations' uptake of HIV testing, but whether the extent of marginalisation challenges individual uptake of HIVST remains under-researched. We aim to explore the perspectives of multiply marginalised cis-gender gay, bisexual and other men who have sex with men (GBMSM) and trans women on whether HIVST might increase their uptake of HIV testing.

### Methods

We reanalysed qualitative interview data from *SELPHI* (the UK's largest HIVST randomised trial) collected between 2017 and 2020 from marginalised populations, defined as people self-identifying as non-heterosexual, transgender, non-White ethnicity and/or with low educational attainment. Thirty-eight interviews with multiply marginalised individuals were thematically examined using the framework method. We specifically focussed on kit usability (a function of the interaction between blood-based HIVST kits and users), perspectives on how HIVST can improve access to HIV testing and suggestions on need-based scale-up of HIVST.

### Results

HIVST kits were considered usable and acceptable by multiply marginalised GBMSM and trans women. The majority of interviewees highlighted multi-levelled barriers to accessing HIV testing services due to structural and social marginalisation. Their multiply marginalised

about the lived experiences of individuals self-identifying as gay, bisexual and other men or transgender people). Depositing our data in public domains will hugely breach compliance with the our study protocol approved by the Research Ethics Committee of University College London. Our anonymised data may be available upon reasonable request with the University College London Research Ethics Committee (contact via Email: ethics@ucl.ac.uk) for researchers who meet the criteria for data access.

**Funding:** This manuscript presents independent research funded by the National Institute for Health and Care Research under the Programme Development Grants (Reference number: NIHR203298). The views expressed in this manuscript are those of the author(s) and not necessarily those of the NHS, the National Institute for Health and Care Research or the Department of Health and Social Care. The funders had no role in study design, data collection and analysis, decision to publish, or preparation of the manuscript.

**Competing interests:** The authors have declared that no competing interests exist.

**Abbreviations:** ABLA, Asian, Black and Latin American; GBMSM, Gay, Bisexual and other Men who have Sex with Men; GCSE, General Certificate of Secondary Education; HIVST, HIV Self-Testing; PrEP, Pre-Exposure Prophylaxis; LGBT, Lesbian, Gay, Bisexual, Transgender; SELPHI, HIV Self-testing Public Health Intervention; STI, Sexually Transmitted Infection.

identities did not impede HIVST uptake but often form motivation to self-test. Three potential roles of HIVST in the HIV testing landscape were identified: (1) alternation of HIVST and facility-based testing, (2) integration of HIVST into sexual health services and (3) substituting facility-based testing with HIVST in the future. Perceived beneficiaries of HIVST included trans communities, individuals with undisclosed sexuality and people with physical disabilities.

## Discussion

HIVST can facilitate marginalised populations' access and uptake of HIV testing by alternating with, integrating into and substituting for facility-based services in the UK. Marginalised identities did not present challenges but rather opportunities for person-centred scale-up of HIVST. Future implementation programmes should ensure equitable access to HIVST among trans people, men unable to disclose their sexuality, and perhaps people with physical disabilities.

## Introduction

The UK's successful HIV epidemic control is not shared equitably among marginalised populations, including people disproportionately facing discrimination or exclusion due to their ethnicity, gender/sexual identity and sexual orientation [1]. Despite an overall decrease in the number of HIV diagnoses among gay, bisexual and other men who have sex with men (GBMSM) in 2022, England reported a 25% increase in new diagnoses among GBMSM with a mixed or other ethnic background [2]. Moreover, the proportion of late HIV diagnoses (defined as CD4 count less than 350 cells per cubic millimetre [3]) among non-White British residents has been increasing since 2020 [2, 4]. These statistics accentuate that people of non-White ethnicity continue to bear disproportionate burdens of HIV, including higher incidence of HIV acquisition and stigma associated with HIV-positive status. Such burdens are often compounded if one also holds other characteristics associated with marginalised communities (e.g. low educational attainment and non-heterosexual) [5]. The HIV-related burdens among minorities with intersecting identities [6, 7] call for urgent action to facilitate their equitable uptake of HIV testing, preventative measures and comprehensive care.

Marginalised populations often face multiple barriers to accessing HIV testing facilities. A great volume of literature has described how stigma and discrimination at interpersonal, institutional and societal levels may deter people from testing for HIV [8–11]. Further, individual disparities in healthcare access may be exacerbated by lower health literacy, disability status (e.g., facing difficulties in visiting clinics), and non-urban residency (e.g., lack of HIV testing facilities in suburban and rural areas) [12–14].

HIV self-testing (HIVST) may improve marginalised populations' uptake of HIV testing [15, 16]. Unlike facility-based testing or HIV self-sampling (which involves posting a sample to a laboratory for processing), HIVST enables a person to use a rapid diagnostic kit for HIV and immediately interpret the result by themselves. Such advantages empower testers to keep confidentiality without relying on test results from overstretched healthcare services. Since 2019, HIVST has been recommended by the World Health Organization [17]. Its feasibility and acceptability in the UK has been established by the HIV Self-Testing Public Health Intervention (SELPHI) trial [18]. Offering free blood-based HIVST kits to 10,135 GBMSM and

trans women enrolled online, SELPHI demonstrated high feasibility and acceptability of HIVST with 95% of kit recipients who filled out surveys reporting having used the self-testing kit [18]. Formative research conducted prior to SELPHI [19–21] identified potential barriers to HIVST use, including concerns about seroconversion, worries about potential harms and a perceived lack of testing support. Qualitative sub-studies including interviews with SELPHI participants (in both pre-COVID and post-COVID contexts where self-testing was widely accepted) have also explored how HIVST empowers GBMSM and trans women to take charge of their health by offering a confidential, convenient and easy-to-use option of HIV testing [22–24].

Decisions around accessing HIV testing (including HIVST) are complex and context-dependent. Individuals may decide to use HIVST in response to perceived testing barriers relevant to their minority identities. For example, Nicholls et al. indicated that some men of Asian, Black and Latin American (ABLA) ethnic backgrounds in SELPHI may prefer HIVST over facility-based testing due to anxiety and discomfort at attending clinics and concerns over implied disclosure of same-sex activity [23]. Qualitative studies from Argentina, the Philippines and the UK also suggest that trans women may seek HIVST for privacy, convenience and alleviation of perceived stigma in healthcare facilities [25–27]. Questions on the dynamics between individuals' extent of marginalisation and their uptake of HIVST remain unanswered. There is a need to understand how HIVST might facilitate increased HIV testing uptake among individuals who may be marginalised due to ethnicity, educational attainment, and gender/sexual identity, especially someone whose identity is impacted by more than one commonly marginalised attribute (often described as a multiply marginalised person) [28, 29]. Understanding such intersectional challenges is key for future HIVST programmes to benefit key populations at risk of HIV in the UK or countries with similar universal health care systems [7, 30]. This paper aims to explore the perspectives of multiply marginalised GBMSM and trans women on whether and how HIVST might increase their uptake of HIV testing.

## Methods

### Study design and rationale

This study applied an interpretive qualitative design by re-analysing personal interviews collected from SELPHI. Throughout SELPHI and its sub-studies, we acknowledged marginalisation as an emerging theme that required a new research inquiry. Reanalysing personal interviews was advantageous for identifying potential patterns or common perspectives across individuals with various attributes of marginalisation [31]. Our secondary data analysis particularly explored how marginalisation (defined below) may determine the perceived usability of testing kits and trust in testing results to inform recommendations on HIVST scale-up for GBMSM and trans people.

### Data source

The SELPHI Qualitative Dataset comprised 86 semi-structured personal interviews from three sub-studies on distinct groups: trans people (n = 20), GBMSM in general (n = 37) and GBMSM of ABLA ethnic backgrounds (n = 29) [32]. These 86 individuals were purposively sampled from all 10,135 SELPHI participants (offered blood-based BioSURE® HIV Self Test Kit) consenting to qualitative sub-studies. All authors (except for IYC) were involved in the data collection of SELPHI and its sub-studies. EJN, PS, TCW and TW conducted personal interviews either online or in-person in English with audio recordings. Each recording was transcribed verbatim, anonymised and de-identified from SELPHI trial participants. The

details of SELPHI's protocol, trial outcomes and the Qualitative Dataset have been published elsewhere [18, 24, 32–34].

## Sampling strategy for data analysis

We purposively sampled interviews from those whose identity is impacted by more than one attribute commonly marginalised in the UK. Informed by British contexts of marginalisation [35, 36], we defined marginalised populations as those who self-identified as LGBT (including cisgender and transgender GBMSM), non-White ethnicity or low levels of formal education (i.e., no higher educational attainment than the General Certificate of Secondary Education (GCSE)). This definition aligned with our purposive sampling per demographic information consistently available across all interviewees (i.e., ethnicity and self-reported highest educational qualifications). Other known factors associated with marginalisation, such as residential location, immigration status and socioeconomic status, were not considered in our sampling framework due to data unavailability. We did not collect data on disability status, but this characteristic emerged as a small number of participants highlighted physical disability as a contributing factor in marginalisation.

Given that all 86 interviewees in the SELPHI Qualitative Dataset were sexual and gender minorities (i.e., GBMSM and trans women as one marginalised attribute), our sampling strategy focussed on those with additional attributes of marginalisation (i.e., non-White ethnicity or low levels of formal education). This resulted in 13 interviewees from the trans sub-study, 15 interviewees from the sub-study on GBMSM in general and 10 interviewees from the ABLA sub-study, respectively. All participants in the ABLA sub-study were non-White GBMSM and most were awarded degrees, so we identified one person with a low level of education and then randomly selected nine ABLA participants to ensure a substantial representation of individuals from diverse ethnic backgrounds. In summary, our purposive sampling for data analysis yielded 38 transcripts from multiply marginalised individuals.

## Data analysis

We adopted the framework method [37] to systematically examine the 38 sampled interviews. IYC accessed the Qualitative Dataset confidentially, familiarised himself with all interview scripts and co-developed an analytical framework with PW. The framework focussed on themes either pertinent to HIVST implementation or underexplored in marginalised populations, such as kit usability, trust in the likely testing results, knowledge about the window period of HIVST and perspectives on how HIVST can improve access to HIV testing [19, 22]. Feasibility is defined as the extent to which interventions can be successfully delivered to target populations (in our case, delivering HIVST to GBMSM and trans people in England and Wales) [38], whereas usability is defined as a function of the interaction between the users and the technology (i.e., self-testing kits) [39]. Knowledge of the window period was assessed by whether interviewees correctly described its meaning and length (estimated at between four and 12 weeks for the blood-based BioSURE® HIV Self Test Kit used in SELPHI [34]). Particular attention was paid to emerging themes, codes or narratives about the potential influence of marginalised identities on interviewees' self-testing experiences.

After analysing the 38 interview transcripts, IYC discussed the findings with AR, FMB, PS, PW and TCW. The team revised the analytical framework and then consulted SELPHI's participant and public involvement group to ensure our analyses and interpretations were ethical, accurate and coherent. We utilised QSR NVivo Version 12 for data analysis and organisation. This study was approved by the University College London Research Ethics Committee (Ref: 24477.001).

## Results

Thirty-eight interviewees included in this study were diverse with regard to gender identity, sexual orientation, ethnicity and HIV testing history upon SELPHI trial enrolment (Table 1). All subsequently used the HIV self-testing kits provided in SELPHI. The majority of interviewees self-identified as cis-gender men, gay/homosexual and of non-White ethnicity. More than a third (15 of 38) reported a low level (i.e., left official education at age 16 or below) of formal educational qualifications.

### Intersectional marginalisation challenged access to HIV testing services

The majority of interviewees highlighted barriers to accessing HIV testing services in the UK due to structural and social marginalisation. They experienced structural marginalisation in accessing HIV testing as public health messaging promoting sexual health and HIV testing seemed disproportionately focussing on cis-gender White MSM. Before using HIVST kits, many also felt marginalised in access to HIV testing because inclusive testing facilities (i.e., user-friendly for gender, sexual or ethnic minorities) were often located in larger urban areas with limited opening hours.

**Table 1. Characteristics of 38 analysed interviewees.**

| Demographic | Category | Count (n = 38) |
|---|---|---|
| Age | 18–25 | 10 |
| | 26–35 | 12 |
| | 36–45 | 9 |
| | 46+ | 7 |
| Gender identity | Cis man | 25 |
| | Trans man | 6 |
| | Trans woman | 7 |
| Sexual orientation | Bisexual | 6 |
| | Heterosexual/Straight | 2 |
| | Homosexual/Gay | 23 |
| | Others/undisclosed | 7 |
| Ethnicity | Asian | 2 |
| | Black | 6 |
| | Latin American | 3 |
| | Mixed | 11 |
| | White | 16 |
| Higher education qualifications | Low* | 15 |
| | Medium** | 14 |
| | High*** | 9 |
| HIV testing history at SELPHI trial enrolment | Less than three months ago | 6 |
| | Three to six months ago | 6 |
| | Seven to 12 months ago | 13 |
| | More than 12 months ago | 7 |
| | Never tested | 6 |

\* General Certificate of Secondary Education (GCSE, leaving official education at age 16) and below

\*\* A-levels or equivalent higher education qualifications

\*\*\* Degree(s) or higher

SELPHI: An HIV Self-Testing Public Health Intervention

Expectations of marginalisation reinforce people's felt stigma of their gender identity, sexuality and ethnicity and thus discourage the utilisation of HIV testing services. Those having multiple minority identities underlined the day-to-day challenges they experienced in accessing sexual health services, including stereotypic assumptions about their sex and gender identity, moral judgments on non-heterosexual relationships and stigma against their sexual practices and HIV. One interviewee shared observations on his Black bisexual peer living in a suburban area:

> *He is African as well, Black African.* [. . .] *He's bisexual as well. But yeah, he does get really scared* [of HIV]. *Even going to get tested, he can't do it locally. . .. He goes either to London somewhere that* [he] *is not going to be* [known by anyone] *because of that stigma of feeling like, 'If I go in there, they* [people in his residential area] *will know. They will not even know that I am either* [HIV] *negative or positive, they are just going to assume I am* [HIV] *positive.*

(27-year-old Black bisexual man, Medium level of education)

Another participant pointed out how intersecting identities (e.g., being a trans man with physical disabilities) challenged his access to sexual health services. He reported needing to make extra efforts to access specific clinics that normalise cis-gender and able-bodied patients by default:

> *I kind of ruminate on things for a long time before I get around to actually get up the courage to do them* [tests for HIV]; *it's not just* [about] *this but like lots of different things. Yeah, there* [are] *just so many different barriers in the way that, yeah, it's hard. A lot of people don't get it,* [they are saying something] *like, 'Just go!' But you are like a perfectly able*[-bodied] *cis gay man. There* [are] *barriers for you but nowhere near as many barriers* [as mine].

(35-year-old trans man with a mixed ethnic background, High level of education)

## Kit usability and marginalisation

All interviewees sampled from SELPHI sub-studies felt that HIV self-testing kits were easy to use. Despite varying levels of educational attainment, only one participant reported difficulties in reading and understanding the instructions printed on the cardboard sleeve inside the kit package. Most participants said that they read the instructions only once and followed them step-by-step to complete testing. Few read the instructions repeatedly and watched video demonstrations, recognising that they were either anxious about HIV or cautious about the multiple steps for self-testing. All felt less anxious and more confident using the kit after completing their first self-test.

Kit usability was not compromised by factors underlying marginalisation. No specific accounts or patterns were identified between participants' perceived kit usability and their ethnic and/or educational backgrounds. Moreover, interviewees' experiences in using the kit and reading testing results were generally positive. No participants reported literacy issues in reading and understanding instructions on self-testing. Most participants appreciated the plain packaging, intuitive design and clear instructions with the kits used in SELPHI:

> *It was really easy especially because you could put the . . . tubing back in the box, and it was lined up with the results. And it would tell you if* [the result] *is here. It's basically that, so it was very easy to read quite quickly what the results were. There wasn't any sort of confusion in it* [the kit], *which is good.*

(22-year-old White trans man, Low level of education)

Nevertheless, kit usability could be compromised by potential manufacturing caveats in interpreting testing results. For instance, despite following the instructions, a participant was unsure if a red mark that emerged at the very bottom of the test strip indicated likely positive results or merely normal blood-reagent reactions:

Interviewer: *The two red lines come in the middle of the white thing, but sometimes there's a thing* [faint red mark appearing at the bottom of the section].

Participant: *Possibly yes*, *if it says that in the instructions*, *I didn't read it properly. I assumed if there were any two lines in that section, it meant positive.* [. . .] *'Is that close to the blue line or not*?*' 'What is close to the blue line*?*' I remember thinking* [about these questions]. *I wasn't sure of the reading* [of my self-testing results] *sometimes.*

(28-year-old Asian gay man, High level of education)

## Trust in testing results and marginalised identities

Irrespective of interviewees' educational attainment and ethnicity, most trusted the negative HIV results arising from self-testing kits. Their trust in self-testing seemed to be strengthened by an overall trust in public health research ('someone else is behind it') and in the UK's regulations on biomedical products in the market.

*I thought just give it a go. Because I said to myself, if it* [the kit] *wasn't accurate, they* [regulators] *probably wouldn't put it on the market, so I give it a go.*

(28-years-old Black gay man, Low level of education)

Two interviewees contended that concerns or disbelief in negative testing results may arise from general anxiety about HIV or perceived higher risks of contracting HIV. When deciding to trust self-testing results or not, they would factor in personal perceptions of risks (e.g., recent condomless sexual intercourse) and subsequent emotive responses. These interviewees also underlined the importance of having accessible educational materials and information about confirmatory testing and further support printed on the kit sleeve, as it can prevent people from feeling isolated or unsure of the next steps.

*I mean it's not the kit's fault.. . .* [Be]*cause if you ask this question* [about trusting testing results] *to someone who's got no anxiety, they're like, I take that 99 per cent sure and I brush off the one per cent* [doubt]. *But if you ask people with anxiety, they will always tell you, I trust that* [testing results] *99 per cent, of the 99 per cent* [accuracy]*, but I trust that one per cent, as well, of inaccuracy.*

(24-year-old White trans woman, Low level of education)

**Knowledge of the window period for HIV self-testing.**   Among those asked about the window period for self-testing, most correctly described the concept and approximate length of the window period as up to 12 weeks. Instructions printed on the kit sleeve, the online demonstration videos and information on SELPHI's official webpage were commonly mentioned as their sources of knowledge. We did not identify any pattern between knowledge of the

window period and interviewees' intersecting identities (e.g., ethnicity and educational attainment). Of the two participants who misunderstood the window period, one belonged to the Low education group while the other had High educational attainment.

### Suggestions on person-centred scale-up of HIVST programme

Interviewees suggested a variety of person-centred ways HIVST programmes can improve access to HIV testing among themselves and other marginalised groups. Here we present interviewees' perceptions of the pivotal roles and priority beneficiaries of HIVST in improving access to, and uptake of HIV testing.

### Key roles of HIVST in the landscape of HIV testing

Depending on personal preferences, sexual practices and the extent of access to HIV testing facilities, interviewees highlighted three key potential roles of HIVST in the UK's landscape of HIV testing services.

1. *Alternate*: Several interviewees planned to utilise both self-testing and facility-based testing, depending on their lifestyles (e.g., moving home or taking holidays) and the availability of appointments at testing facilities at various time points. They may utilise self-testing and traditional HIV testing services alternately. HIVST may be prioritised when people experience inconvenience in accessing sexual health clinics, whereas sexual health clinics could be prioritised when people need in-person counselling or other types of sexual health services, such as hormone injections, screening for sexually transmitted infections (STIs) or HIV PrEP prescription.

> *Before I discovered the SELPHI* [HIVST] *kit, I always used to go to the doctor. But since I've started the SELPHI kit* [. . .], *I go SELPHI kit, doctor, SELPHI kit, doctor, SELPHI kit, doctor. That depends* [on] *how scared I am of the situation.*

(24-year-old White trans woman, Low level of education)

2. *Integrate*: A few interviewees preferred HIVST to be offered at HIV testing facilities as a walk-in option for testing. They would like to conduct unsupervised self-testing on their own to retain privacy (e.g., in a private room at premises) while having an opportunity for consultation as necessary. Specifically, if self-testing kits yielded inconclusive or likely positive results, users could immediately seek confirmatory testing and professional support from healthcare providers at premises.

> *I tell you what the ultimate would be, is to be able to walk into a clinic and use these 15-minute tests. I think that's the ultimate.* [. . .] *Without the time lag, but then maybe with some support maybe. I guess it's all about choice, isn't it? I guess you want to have the option of both.* [Be]*cause now I think it would be nice just to have the self-test, but only because I think I'm negative. And if I thought I was positive, oh, I don't know.*

(35-years-old Black bisexual man, High level of education)

3. *Substitute*: Regardless of availability, some interviewees preferred to completely switch to HIVST instead of facility-based services to overcome institutional barriers to testing, such as geographical disparity, little availability of appointments and inconvenient access to premises. Ensuring personal privacy but with linkage to professional support was often mentioned by this group. They suggested that the self-testing kit should be accompanied by lay-person-

friendly information on seeking further support when necessary. Such information can be provided in multiple ways based on users' preferences, including free hotlines, online chatbots and self-registration at the webpage of local sexual health services for confirmatory testing.

**Priority beneficiaries of HIVST.** Participants also underscored three marginalised populations that they felt could most benefit from self-testing, given that these populations were considered neglected at HIV testing facilities. Firstly, many trans interviewees believed that HIVST could improve their sexual well-being as many services were deemed unfriendly to trans communities. Secondly, self-testing can empower people unable to open up about their sexual orientation, including (but not limited to) younger and ABLA MSM living in heteronormative environments (e.g., those living in extended multi-generational families). This group can utilise HIVST to fulfil HIV testing needs while retaining their privacy. Thirdly, some interviewees pointed out that HIVST can greatly empower people with physical disabilities, who were considered especially marginalised in facility-based services.

*Like, if you depend on other people to get places, that can leave you a lot more isolated and vulnerable.* [For] *anyone receiving agency care, the carer is so hit and miss, and there is a lot of judgement sometimes that people place on people* [with disabilities] *in terms of accessing* [sexual health] *services. Like,* [if I ask people:] *'Would you take me to the* [sexual health] *clinic'?* [And they answer:] *'What do you need to go there for?' Just rule it* [sexual health] *out* [by] *some people.*

(48-year-old trans man with a mixed ethnic background)

## Discussion

Examining the accounts of 38 HIVST kit users in SELPHI, we provide clear evidence (summarised in Table 2) that blood-based HIV self-testing kits are usable and acceptable among multiply marginalised GBMSM and trans women of non-White ethnicity and/ or low educational attainment. Their multiply marginalised identities were no impediment to uptake or kit usage and often formed part of the motivation to test using HIVST, especially for trans individuals, those with undisclosed sexuality or people with physical disabilities. For many the privacy afforded by HIVST contributes a large part of the motivation to test using the kit. We also underline three perceived roles of HIVST: alternation of HIVST and facility-based testing, integration into sexual health clinics and perhaps sole use of HIVST for future testing.

Our analysis demonstrates high usability of HIVST kits among SELPHI participants with intersecting identities. Consistent with the findings from previous SELPHI research, this study highlights the resilience and self-efficacy of marginalised populations in utilising HIVST to improve their well-being. While some literature may attribute individuals' low HIV testing rates to marginalised identities (e.g., being migrants or ethnic minorities) [40, 41], our findings suggest that a person-centred and needs-based design of HIVST programmes may facilitate marginalised populations' uptake of self-testing and help break intersectional barriers to access. Building upon SELPHI's experience, further evaluation of roll-out is warranted to understand the real-world barriers and facilitators to implementing HIVST in countries with universal healthcare systems, so as to minimise unintended consequences [21, 42, 43].

Our findings highlighted various potential modalities for implementing HIVST programmes based on the diverse needs of potential testers. Both the alternate and integrated roles of HIVST in HIV testing services were consistent with other studies in high-income

**Table 2. Summary of key findings.**

| Theme | Key finding |
|---|---|
| **Context:** Multiply marginalised populations face continual challenges accessing HIV testing services | Structural and social marginalisation reinforced people's felt stigma regarding their gender identity, sexuality, or ethnicity, discouraging the utilisation of HIV testing (and other sexual health services) among multiply marginalised populations. |
| **Experience:** People with marginalised identities can utilise HIVST to enhance their HIV testing by ensuring: | |
| Kit usability | Factors underlying marginalisation did not impact the perceived usability of HIVST kits. |
| Trust in testing results | Factors underlying marginalisation did not undermine trust in the negative results of HIVST kits. |
| Knowledge about the window period of HIVST | Factors underlying marginalisation did not undermine people's capacity to understand the length of the window period of HIVST kits. |
| **Suggestion:** Key considerations for person-centred HIVST programmes: | |
| Three potential roles of HIVST in HIV testing services | Alternation between HIVST and facility-based testing |
| | Integration of HIVST into sexual health services |
| | Possible substitution of facility-based testing with HIVST |
| Three priority beneficiaries of HIVST | Transgender communities |
| | Individuals with undisclosed sexuality |
| | People with physical disabilities (limited data) |

HIVST: HIV Self-Testing

settings as they may alleviate the increasing workload of over-stretched sexual health services [42, 44, 45]. The accounts of standalone use of HIVST emphasise that HIVST should be offered as an available option if countries aim to improve equity, accessibility and scalability of HIV testing. While some researchers and policymakers warn that people using home-based HIVST may miss opportunities for immediate support and adherence to regular HIV testing [42, 46], our study reveals that both information printed on self-testing kits (including quick response codes to video demonstration) can provide users with immediate support and linkage to confirmatory testing. Moreover, those who face difficulties in either facility-based testing (e.g., perceived structural barriers and stigma) or self-sampling (e.g., worries about confidentiality and delayed testing results), can benefit from the privacy, efficiency and self-guided support of HIVST. As individuals' needs and perceived levels of social support vary, implementors may take a pluralistic approach in designing HIVST programmes. An HIVST programme with non-facility-oriented support (e.g., hotlines or telemedicine consultation) [47] should be taken into consideration, so the testing needs of marginalised populations can be met.

Our findings imply that self-testing could empower people with physical disabilities by fulfilling their underserved needs for HIV and STI testing. The needs for HIV prevention and care among people with disabilities are under addressed worldwide [48, 49]. Except for Middleton et al.'s exploration of STI self-sampling among people with learning disabilities [50], to our knowledge, there is no published research investigating the acceptability, usability, or unmet needs for testing of people with physical disabilities. Our findings imply that people with physical disabilities could be potential beneficiaries of HIVST. To ensure 'no one left behind' [51] in the roll-out of HIVST programmes, we encourage researchers to include people with physical disabilities in future studies on HIV testing and prevention.

## Limitations

Our study contributes to the literature on HIV testing by highlighting the capabilities of marginalised populations to utilise HIVST in the context of a randomised trial in a high-income setting. However, several limitations exist. Firstly, our study is of limited generalisability as it only analyses participants with multiple intersecting identities in the SELPHI randomised trial. Subject to the limited number of interviewees and the context of SELPHI as a randomised trial, our interpretation cannot generalise or represent perspectives on HIVST among all marginalised communities residing in England and Wales. It is worth noting that, even in the UK, no epidemiological data on HIV testing uptake and/or outcomes are routinely reported for multiply marginalised populations.

Secondly, the high usability and trust in HIVST reported by interviewees may arise from the specific design of SELPHI, social desirability bias and selection bias. Participants have regularly received information on HIVST, HIV prevention and care since their trial enrolment, so it is unsurprising that they may have better knowledge about HIVST. As most data were collected during and shortly after the trial period, participants may have expressed opinions favouring the research team. Also, those willing to be interviewed may be more satisfied than their counterparts.

Thirdly, as our study was not designed to investigate issues related to physical disabilities, our proposed potential benefits of HIVST on people with physical disabilities should be cautiously interpreted. Nor were the interview data originally collected to explore the dynamics of intersectional marginalisation and HIVST. Specifically, perspectives on disabled people should be understood as interviewees' conjecture rather than actual lived experiences. Moreover, it was impossible to verify such interview accounts because no data on participants' disability status were collected in SELPHI and its qualitative sub-studies.

## Conclusions

Our study identifies the promising roles of HIVST in facilitating access to HIV testing for marginalised populations by alternating with, integrating into and substituting for facility-based services. Marginalised identities were not challenges but opportunities for person-centred scale-up of HIVST. Future programmes should ensure the rights to accessing HIVST of socially marginalised populations, including trans people, men unable to disclose their sexuality, and perhaps people with physical disabilities.

## Acknowledgments

We would like to acknowledge the invaluable support of our community advisory group, including its co-chairs Roger Pebody and Roy Trevelion. We sincerely express our gratitude to all participants in SELPHI and its sub-studies.

## Author Contributions

**Conceptualization:** Fiona M. Burns, T. Charles Witzel, Emily Jay Nicholls, Alison J. Rodger, Peter Weatherburn.

**Data curation:** Talen Wright, Phil Samba, T. Charles Witzel, Emily Jay Nicholls.

**Formal analysis:** Isaac Yen-Hao Chu.

**Funding acquisition:** T. Charles Witzel, Alison J. Rodger.

**Investigation:** Isaac Yen-Hao Chu.

**Methodology:** Isaac Yen-Hao Chu, T. Charles Witzel, Peter Weatherburn.

**Project administration:** Isaac Yen-Hao Chu, Alison J. Rodger.

**Software:** Isaac Yen-Hao Chu.

**Supervision:** Fiona M. Burns, T. Charles Witzel, Alison J. Rodger, Peter Weatherburn.

**Visualization:** Isaac Yen-Hao Chu.

**Writing – original draft:** Isaac Yen-Hao Chu.

**Writing – review & editing:** Isaac Yen-Hao Chu, Fiona M. Burns, Talen Wright, Phil Samba, T. Charles Witzel, Emily Jay Nicholls, Leanne McCabe, Andrew Phillips, Sheena McCormack, Alison J. Rodger, Peter Weatherburn.

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
