## [Decision Letter · Decision Letter 0]

14 Jun 2024

PONE-D-24-15187How can HIV self-testing facilitate increased access to HIV testing among multiply marginalised populations? Perspectives from GBMSM and trans women in England and WalesPLOS ONE

Dear Dr. Yen-Hao Chu,

Thank you for submitting your manuscript to PLOS ONE. After careful consideration, we feel that it has merit but does not fully meet PLOS ONE’s publication criteria as it currently stands. Therefore, we invite you to submit a revised version of the manuscript that addresses the points raised during the review process.

We look forward to receiving your revised manuscript.

Kind regards,

Praveen Suthar, MPH

Academic Editor

PLOS ONE

“This manuscript presents independent research funded by the National Institute for Health and Care Research under the Programme Development Grants (Reference number: NIHR203298). The views expressed in this manuscript are those of the author(s) and not necessarily those of the National Institute for Health and Care Research or the Department of Health and Social Care.”

3. In the online submission form, you indicated that [Anonymised data are available upon reasonable request.].

Additional Editor Comments:

This is well-written manuscript. However, I have some comments and suggestions for improvement.

Introduction:

1. The introduction should define "marginalized" in the context of their research. This would enable the reader to define the study population as early as the introduction. What qualifies as marginalized?

2. The introduction should contain local and international studies on qualitative assessment of HIVST among the same or similar population. This would enhance the background of the study and contextualize the current landscape on HIVST.

3. What does "multiply" mean in page 7 line 113?

Methods:

1. Kindly define the study design used in the qualitative investigation and the reason for choosing such design.

2. How did you select the study population? It is not clear how sampling procedure was done.

3. It is better to have the methods section with different subheadings in order to clearly delineate one section from another.

4. I reckoned the the authors re-analyzed the transcripts from a a trial. However, it might be better elaborate more on how these transcripts were collected and re-analyzed.

5. The type of HIV self-testing was not clearly stated.

Results:

1. The results were well-written and structured. However, it might be better if a summary of the main themes using a table or figure should be included in order to clear present the results of the analysis.

2. Page 11, line 176: you mentioned "most". How many were this "most"? Kindly qualify by including (n=X) in the statement.

3. I am having difficulty in following the results as to whether this answered the main research question of: to explore the perspectives of multiply marginalized GBMSM and trans women on whether and how HIVST might increase their uptake of HIV testing.

Discussion and Conclusion:

1. I want to see how the discussion and conclusion will navigate based on the concept map generated from the study results and how these themes relate to another. I am having difficulty navigating these sections.

2. Are there epidemiological data of marginalized GBMSM and trans women in the country? I suggest to include these data and how the results of the study may help in easing the access of HIVST among these identified sets of population.

Reviewers' comments:

Reviewer's Responses to Questions

**Comments to the Author**

1. Is the manuscript technically sound, and do the data support the conclusions?

Reviewer #1: Yes

2. Has the statistical analysis been performed appropriately and rigorously? 

Reviewer #1: N/A

3. Have the authors made all data underlying the findings in their manuscript fully available?

Reviewer #1: No

4. Is the manuscript presented in an intelligible fashion and written in standard English?

Reviewer #1: Yes

5. Review Comments to the Author

Reviewer #1: This is well-written manuscript. However, I have some comments and suggestions for improvement.

Introduction:

1. The introduction should define "marginalized" in the context of their research. This would enable the reader to define the study population as early as the introduction. What qualifies as marginalized?

2. The introduction should contain local and international studies on qualitative assessment of HIVST among the same or smilar population. This would enhance the background of the study and contextualize the current landscape on HIVST.

3. What does "multiply" mean in page 7 line 113?

Methods:

1. Kindly define the study design used in the qualitative investigation and the reason for choosing such design.

2. How did you select the study population? It is not clear how sampling procedure was done.

3. It is better to have the methods section with different subheadings in order to clearly delineate one section from another.

4. I reckoned the the authors re-analyzed the transcripts from a a trial. However, it might be better elaborate more on how these transcripts were collected and re-analyzed.

5. The type of HIV self-testing was not clearly stated.

Results:

1. The results were well-written and structured. However, it might be better if a summary of the main themes using a table or figure should be included in order to clear present the results of the analysis.

2. Page 11, line 176: you mentioned "most". How many were this "most"? Kindly qualify by including (n=X) in the statement.

3. I am having difficulty in following the results as to whether this answered the main research question of: to explore the perspectives of multiply marginalised GBMSM and trans women on whether and how HIVST might increase their uptake of HIV testing.

Discussion and Conclusion:

1. I want to see how the discussion and conclusion will navigate based on the concept map generated from the study results and how these themes relate to another. I am having difficulty navigating these sections.

2. Are there epidemiological data of arginalised GBMSM and trans women in the country? I suggest to include these data and how the results of the study may help in easing the access of HIVST among these identified sets of population.

6. PLOS authors have the option to publish the peer review history of their article (what does this mean?). If published, this will include your full peer review and any attached files.

Reviewer #1: No

---

## [Author Response · Author response to Decision Letter 0]

17 Aug 2024

Dear Editor:

Thank you very much for your advice. We provide more information on how our manuscript has met PLOS ONE’s requirements in the attached Cover Letter.

Dear Reviewers:

Please find our itemised responses to your requests/comments in the file named Response to Reviewer. All line numbers and pages indicated below are based on the file named 'Manuscript' (i.e., the revised manuscript without track changes).

Thank you for your patience and kind consideration. We look forward to hearing from you soon.

Best regards,

Isaac Yen-Hao Chu

---

## [Decision Letter · Decision Letter 1]

4 Oct 2024

PONE-D-24-15187R1How can HIV self-testing facilitate increased access to HIV testing among multiply marginalised populations? Perspectives from GBMSM and trans women in England and WalesPLOS ONE

Dear Dr. Yen-Hao Chu,

Thank you for submitting your manuscript to PLOS ONE. After careful consideration, we feel that it has merit but does not fully meet PLOS ONE’s publication criteria as it currently stands. Therefore, we invite you to submit a revised version of the manuscript that addresses the points raised during the review process.

 The revised manuscript is overall of excellent quality and the authors have done a great job in addressing the reviewers' feedback. However, I agree with the reviewer that "pragmatic qualitative approach" is not a sufficient description of the study design. This is a minor issues that I'm sure the authors can address quickly.

We look forward to receiving your revised manuscript.

Kind regards,

Daniel Demant, PhD, MPH, GradCertHEd, BAppSocSc

Academic Editor

PLOS ONE

Journal Requirements:

Reviewers' comments:

Reviewer's Responses to Questions

**Comments to the Author**

1. If the authors have adequately addressed your comments raised in a previous round of review and you feel that this manuscript is now acceptable for publication, you may indicate that here to bypass the “Comments to the Author” section, enter your conflict of interest statement in the “Confidential to Editor” section, and submit your "Accept" recommendation.

Reviewer #1: (No Response)

2. Is the manuscript technically sound, and do the data support the conclusions?

Reviewer #1: Yes

3. Has the statistical analysis been performed appropriately and rigorously? 

Reviewer #1: N/A

4. Have the authors made all data underlying the findings in their manuscript fully available?

Reviewer #1: No

5. Is the manuscript presented in an intelligible fashion and written in standard English?

Reviewer #1: Yes

6. Review Comments to the Author

Reviewer #1: I congratulate the authors for a great job in revising the manuscript. However, I still have some comments:

1. The term "pragmatic qualitative design" is not a standard or widely recognized qualitative research design. To ensure clarity and adherence to qualitative research methodologies, it would be beneficial for the authors to explicitly state the specific qualitative research design employed in the study (e.g., phenomenology, grounded theory, case study, etc.). This will not only provide methodological transparency but also align the study with established research frameworks in qualitative inquiry.

7. PLOS authors have the option to publish the peer review history of their article (what does this mean?). If published, this will include your full peer review and any attached files.

Reviewer #1: No

---

## [Author Response · Author response to Decision Letter 1]

9 Oct 2024

Dear Reviewers:

Please find our responses to your comment as follows. The line number and page indicated are based on the file named 'Manuscript' (i.e., the revised manuscript without track changes).

Question 1: The term "pragmatic qualitative design" is not a standard or widely recognized qualitative research design. To ensure clarity and adherence to qualitative research methodologies, it would be beneficial for the authors to explicitly state the specific qualitative research design employed in the study (e.g., phenomenology, grounded theory, case study, etc.). This will not only provide methodological transparency but also align the study with established research frameworks in qualitative inquiry.

Answer 1: We appreciate your advice on clarifying the qualitative research design applied in this study. To explicit state our study design, we have revised the sentence on Line 118, Page 8 as 'This study applied an interpretive research design by re-analysing personal interviews collected from SELPHI.'

Additional revision made by authors:

• We have ensured that all in-text quotations in the Results sections are correctly formatted.

• We have revised the References section to ensure that all references are aligned with PLOS One’s reference style.

• We have updated the corresponding email of IYC.

---

## [Editor Report · Decision Letter 2]

16 Oct 2024

How can HIV self-testing facilitate increased access to HIV testing among multiply marginalised populations? Perspectives from GBMSM and trans women in England and Wales

PONE-D-24-15187R2

Dear Dr. Chu,

We’re pleased to inform you that your manuscript has been judged scientifically suitable for publication and will be formally accepted for publication once it meets all outstanding technical requirements.

Kind regards,

Daniel Demant, PhD, MPH, GradCertHEd, BAppSocSc

Academic Editor

PLOS ONE
---

## [Editor Report · Acceptance letter]

20 Oct 2024

PONE-D-24-15187R2 

PLOS ONE

Dear Dr. Chu, 

I'm pleased to inform you that your manuscript has been deemed suitable for publication in PLOS ONE. Congratulations! Your manuscript is now being handed over to our production team.

Kind regards, 

on behalf of

Dr. Daniel Demant 

Academic Editor

PLOS ONE